# Increased signal diversity/complexity of spontaneous EEG, but not evoked EEG responses, in ketamine-induced psychedelic state in humans

Nadine Farnes[1], Bjørn E. Juel[1], André S. Nilsen[1], Luis G. Romundstad[2], Johan F. Storm[1]*

1 Brain Signaling Group, Department of Molecular Medicine, University of Oslo, Oslo, Norway, 2 Department of Anaesthesia, and Intervention Centre, Oslo University Hospital, Oslo, Norway

* j.f.storm@medisin.uio.no

**Data Availability Statement:** Anonymized pre-processed data for spontaneous and TMS-evoked

## Abstract

How and to what extent electrical brain activity reflects pharmacologically altered states and contents of consciousness, is not well understood. Therefore, we investigated whether measures of evoked and spontaneous electroencephalographic (EEG) signal diversity are altered by sub-anaesthetic levels of ketamine compared to normal wakefulness, and how these measures relate to subjective experience. High-density 62-channel EEG was used to record spontaneous brain activity and responses evoked by transcranial magnetic stimulation (TMS) in 10 healthy volunteers before and during administration of sub-anaesthetic doses of ketamine in an open-label within-subject design. Evoked signal diversity was assessed using the perturbational complexity index (PCI), calculated from EEG responses to TMS perturbations. Signal diversity of spontaneous EEG, with eyes open and eyes closed, was assessed by Lempel Ziv complexity (LZc), amplitude coalition entropy (ACE), and synchrony coalition entropy (SCE). Although no significant difference was found in TMS-evoked complexity (PCI) between the sub-anaesthetic ketamine condition and normal wakefulness, all measures of spontaneous EEG signal diversity (LZc, ACE, SCE) showed significantly increased values in the sub-anaesthetic ketamine condition. This increase in signal diversity correlated with subjective assessment of altered states of consciousness. Moreover, spontaneous signal diversity was significantly higher when participants had eyes open compared to eyes closed, both during normal wakefulness and during influence of sub-anaesthetic ketamine. The results suggest that PCI and spontaneous signal diversity may reflect distinct, complementary aspects of changes in brain properties related to altered states of consciousness: the brain's capacity for information integration, assessed by PCI, might be indicative of the brain's ability to sustain consciousness, while spontaneous complexity, as measured by EEG signal diversity, may be indicative of the complexity of conscious *content*. Thus, sub-anaesthetic ketamine may increase the complexity of the conscious content and the brain activity underlying it, while the *level* or general capacity for consciousness remains largely unaffected.

EEG is available on the Dryad repository: https://doi.org/10.5061/dryad.j9kd51c9q.

**Funding:** This study was supported by the European Union's Horizon 2020 research and innovation programme under grant agreement 7202070 (Human Brain Project (HBP)) and the Norwegian Research Council (NRC: 262950/F20 and 214079/F20) to author JFS. The funders had no role in study design, data collection and analysis, decision to publish, or preparation of the manuscript.

**Competing interests:** The authors have declared that no competing interests exist.

## Introduction

Understanding the nature and mechanisms of consciousness is widely regarded as one of the deepest unsolved problems of neuroscience and science in general [1–6]. To advance the field it is crucial to develop ways to measure alterations in states and content of consciousness. To this end, general anaesthetic drugs are among the most powerful interventional tools as they allow quite specific and graded manipulation of the state of consciousness [7], without suppressing many lower-level brain functions [5, 8, 9], and have been crucial for a series of recent discoveries of brain function. These include changes in network activity and connectivity in the anaesthetized state compared to the normal wakeful state using non-invasive neural measures such as fMRI, TMS and EEG [7, 10].

One line of observations and theoretical developments suggests that the brain must be both informationally integrated and differentiated to support consciousness [5, 10, 11]. This idea can to some extent be tested experimentally, albeit indirectly, in biological systems by methods such as the perturbational complexity index [PCI; 12]. PCI is assumed to assess the brain's current capacity for sustaining consciousness, by roughly indexing its potential for information differentiation and integration. PCI is obtained by perturbing a small part of the cerebral cortex with transcranial magnetic stimulation (TMS) and measuring the resulting spatiotemporal responses in large parts of the cortex with high-density electroencephalography (EEG). Then, by quantifying how compressible the spatio-temporal EEG patterns are, one can estimate the Kolmogorov complexity of the TMS-evoked activity. Thus, PCI is a measure of the global, spatio-temporal complexity of evoked, cortical responses to a local perturbation. This is thought to reflect the structural and functional complexity of the underlying system, including both its interconnectedness (integration), and the diversity of its available states of activity (differentiation).

So far, PCI has mainly been used to assess the brain's capacity for complex dynamics in states that appear to differ drastically in terms of consciousness. The results indicate that PCI scores are high during wakefulness, REM sleep (with high frequency of vivid dreams), in ketamine anaesthesia (which can cause long-lasting, vivid hallucinations or dreams), and in locked-in syndrome (a condition of full paralysis, but with intact consciousness) [12, 13]. Conversely, PCI scores are typically low in states of general anaesthesia (other than with ketamine), unresponsive wakefulness syndrome, coma, and deep, NREM sleep, which is assumed to be largely dreamless [12–15]. Thus, reduced PCI scores, indicating reduced integration and/or differentiation, seems to correlate well with reduced capacity for phenomenological experiences (consciousness) relative to normal wakefulness [although dreams or dream-like experiences have been reported following both NREM sleep and general anaesthesia; 12]. For these reasons, PCI is suggested to be a highly promising general index of the brain's capacity for sustaining wake-like complex conscious experiences [4, 14].

While PCI aims to capture both neural differentiation and integration, recent studies have indicated that measures that solely capture differentiation in recordings of spontaneous brain activity, seems to vary with brain state in parallel with PCI [10, 16]. Here, we label such measures of differentiation as "signal diversity measures". Thus, a signal diversity measure quantifies the variation of neural signals in a given physiological brain state and provides an estimate of the differentiation in the signal. In these studies, three measures of signal diversity were successfully applied to brain activity recordings; Lempel-Ziv complexity (LZc), amplitude coalition entropy (ACE), and synchrony coalition entropy (SCE). These measures capture how "disordered", i.e. differentiated, the spatio-temporal signal recorded from spontaneous brain activity is. This can for example be quantified by calculating the signal's compressibility using an adapted Lempel-Ziv compression algorithm or one of the many available entropy measures

(e.g. coalition entropy [10]). While these measures behave similarly as PCI (e.g. decreasing in general anaesthesia and deep sleep) they also show counterintuitive results under some conditions. For example, signal diversity of spontaneous magnetoencephalography (MEG) recordings increased in the psychedelic state induced by LSD, psilocybin, or sub-anaesthetic doses of ketamine, compared to normal wakefulness. This suggests that these psychedelic states are associated with an increase in neural differentiation [17].

While these findings seem to be in contrast with the drop in signal diversity typically seen when consciousness appears to be lost, they do not necessarily yield support to speculations that psychedelics can produce a 'higher' or 'richer' state of consciousness than normal wakefulness [18]. First of all, the use of unidimensional measures to quantify consciousness has been criticized [19], suggesting that the notion of 'higher' or 'lower' levels of consciousness may be misleading. Secondly, even if unidimensional measures capture some relevant property like degree or level of consciousness, some theoretical considerations emphasize that both integration and differentiation are necessary to sustain consciousness [2, 5]. Thus, according to these theories, an increase in signal diversity alone is insufficient evidence for a heightened level of consciousness. Finally, the signal diversity measures are calculated from recordings of spontaneous brain activity. Thus, it is possible that the observed increase is due to factors outside the brain regions relevant for consciousness. An extreme example could be that every neuron in the brain was driven by individual, independent noise generators. In such a situation, signal diversity would be maximal, but it seems unlikely that such a brain would have the capacity to support consciousness. In order to overcome some of these difficulties, one can use a perturbational approach e.g. by measuring PCI [12, 14, 20]. PCI has to our knowledge not previously been tested in any purely psychedelic state.

Among the commonly used general anaesthetics, the dissociative anaesthetic drug ketamine stands out as quite unique, both in terms of cellular mechanisms, electrophysiological patterns, and behavioural effects [21–24]. At high doses, sufficient for surgical general anaesthesia, ketamine induces an unresponsive, dissociated state, with sedation, analgesia and amnesia, but with strong inner experiences in the form of vivid dreams [13, 25]. Unlike "typical" general anaesthetics, ketamine also preserves wake-like levels of PCI [13], and alternating wake-like and low levels of signal diversity [5]. At low sub-anaesthetic doses, ketamine has distinctive dissociative and hallucinogenic effects [9, 10], with increased levels of signal diversity similar to that measured in LSD and psilocybin induced psychedelic states [17, 22]. Given these properties, ketamine is an excellent tool for pharmacological manipulation of consciousness, with dose-dependent qualitative and quantitative effects on electrophysiological properties, as well as on reported conscious experience and behaviour.

Our main aim in this study was to investigate whether PCI increases in the psychedelic state induced by sub-anaesthetic levels of ketamine, compared to normal wakefulness. In addition, we wanted to test whether we can replicate reported increases in electrophysiological signal diversity during sub-anaesthetic ketamine [17, 22]. Finally, we aimed to investigate the relationships between subjective reports from the psychedelic state, on one hand, and both PCI and measures of spontaneous signal diversity on the other. Given that previous results from the literature regarding signal diversity replicate, and an assumption that PCI and signal diversity measures covary in healthy humans, we hypothesized that PCI would increase in the psychedelic state compared to the normal wakefulness.

## Methods

This study was approved by the Norwegian Regional Committees for Medical and Health Research Ethics (2015/1520/REK Sør-Øst A) and written informed consent was obtained from

all participants before the start of the study. Participants were recruited by posters at the university campus and participants received financial compensation for participation in the study.

## Participants

We recruited 34 participants. Participants were excluded if they fulfilled any of the following criteria: (1) under the age of 18, (2) incompatibility with MRI scanning (metal or electric implants, pregnancy, breastfeeding, reduced kidney function and claustrophobia), (3) incompatibility with TMS administration (recent loss of consciousness caused by head injury or epilepsy), (4) incompatibility with ketamine administration (somatic diseases, previous or present neurological or psychiatric illnesses, psychiatric illnesses in family members, medication or allergies that could interact with ketamine, substance abuse, recent or regular drug use, previous adverse reaction to drugs, or needle phobia), (5) difficulty finding suitable resting motor threshold for TMS (RMT, i.e. the stimulation intensity at which 50% of TMS pulses over the optimal spot in primary motor cortex generate twitches in the pollicis brevis (thumb) muscle as recorded with surface electromyography [26]), and, (6) low TMS evoked potential (TEP) quality (muscle artefacts and peak to peak amplitude less than 10 μV). The most common reason for exclusion being difficulty finding RMT or low signal to noise ratio in the TEPs. Based on this, 10 participants (7 men and 3 women, median age 27.5, range: 21–44 years old) were included and completed the study (see S1 Fig for a flowchart of participation of the study).

## Procedure

All experimental procedures were carried out at the Intervention Centre at Oslo University Hospital. Before the main experiment, spontaneous EEG and TMS evoked EEG responses were recorded without ketamine (day 1) to assess participants' TEPs to ensure that only participants with strong responses to TMS completed the main experiment (day 2), about 6 days after day 1 (median 6 days; range: 3–23 days). Two TMS-EEG and spontaneous EEG sessions were completed on day 2; baseline and intervention involving administration of sub-anaesthetic doses of ketamine (see S1 Fig). On day 2, we first found the RMT and searched for a suitable target location for TMS stimulation for the main experiment. Thereafter, 300 TMS-EEG trials were recorded while the participants were in a restful, wakeful state with their eyes open. In addition, two 2-minute segments of spontaneous EEG were recorded—first with eyes open and then with eyes closed. The recordings for the eyes-open conditions were done with the lights in the operating theater slightly dimmed. Thereafter, participants were given an intravenous infusion of sub-anaesthetic doses of ketamine. When the participant reached a state in which they reported a noticeable effect of the drug, we measured the RMT again [27], and adjusted the TMS stimulation intensity to the same percentage of the RMT as was used during the wakeful state. We then delivered another 300 TMS pulses, while keeping the stimulation target the same as before ketamine administration. In addition, spontaneous EEG was recorded under the same conditions, once again asking the participants first to keep their eyes open and then to have their eyes closed. The TMS-EEG measurements were only done with eyes open to keep the time spent under the effect of ketamine short, as the required measurements take at least 10 minutes and the ethical approval only allowed one set of TMS-EEG measurements per condition (normal wakefulness and ketamine).

During the stimulation period the participants wore earphones with a noise masking sound, made from randomly scrambled audio recordings of TMS clicks, to reduce auditory potentials evoked by the TMS clicks [28]. The noise masking sound was adjusted so that the TMS click could not be heard but was never so loud that the sound became uncomfortable for

the participant. Throughout the procedure, participants were lying down and had their head fixed on a vacuum pillow to ensure stability during stimulation.

**Identifying target area for TMS stimulation.**    On day 1, based on each participant's MR image, a suitable target point for stimulation was identified within the parietal (Brodmann Areas: BA 7, BA 5) or prefrontal cortex (BA 6), similar to the procedures used in [14] and [12]. If there were no large muscular or magnetic artefact lasting over 10 ms visible in the EEG response to single pulses, 20–30 pulses were given, and the resulting TEP was examined online. If the TEP amplitude was below 10 µV peak-to-peak (using an average reference) within the first 50ms, we tried to improve the TEP signal by increasing the intensity of stimulation. If this did not improve the TEP or introduced more artefacts, we adjusted coil rotation or position. The target point for stimulation was accepted if a non-artefactual TEP, with a peak-to-peak amplitude equal to or exceeding 10 µV in the channels near the stimulation site in the first 50 ms after stimulation, was observed in the averaged TEP signal following 30 consecutive single pulses. When the stimulation area was found, 300 single pulses were given with a random jittering inter-stimulus interval [range: 1.7–2.3 seconds, 4]. One participant was stimulated in BA 4 because this was the only area without large muscle artefacts (See S1 Table for an overview of the TMS targets and stimulation parameters for all participants). Navigation of the coil position and angle relative to the participant's brain was used to minimize the deviation from the set target point position (Neuro-navigation with PowerMag View). To reduce the distance between the centre of the coil and the target point, care was taken to keep the centre of the coil positioned as tangentially to the scalp as possible [29]. The procedure for finding an area of stimulation was the same for day 1 and day 2.

**Ketamine administration.**    Participants were told to abstain from food 6 hours before the first recording session started and from drinking 2 hours before ketamine administration. Racemic ketamine 10 mg/ml (Ketalar®, Pfizer AS, Lysaker, Norway) was administered by an anaesthesiologist or a nurse anaesthetist by continuous intravenous infusion using an infusion pump (Braun®) in increasing steps from 0.1 to 1.0 mg/kg/h, increasing with 0.1 mg/kg/h every fifth minute. The participants were asked to report when they felt they had a possible drug effect and when they were certain that they had an effect. When both the anaesthesiologist and the participant were certain that the participant had an effect of the drug, the continuous ketamine infusion rate was stabilized to maintain the subjective effect (B. Braun Perfusor Space, B. Braun Melsungen AG, Melsungen, Germany). The median continuous infusion was 0.7 mg/kg/hr (range: 0.5 to 1.0 mg/kg/hr) for each participant, producing psychotomimetic effects without loss of consciousness [30]. See supporting information for maintained, continuous dose, total dose received for each participant, and dosage steps over time. Participants' pulse oximetry and heart rate was continuously monitored during the ketamine administration by the anaesthesiologist or the nurse anaesthetist. Median administration time was 43.5 minutes (range: 37–73 minutes). Participants could leave the hospital facilities after the anaesthesiologist had checked to ensure that the effects of the drug had subsided (approximately 2 hours after discontinuation of ketamine administration). A follow-up email was sent to the participants more than a week after the finished experiment to check their wellbeing. One participant experienced nausea right after the end of the experiment, but no other unwanted adverse effects of low-dose ketamine was reported.

**Psychedelic assessment: 11D-ASC.**    To assess the psychedelic effects of ketamine relative to the pre-ketamine condition, participants retrospectively rated the content of their experience using an altered states of consciousness questionnaire: an extended, 11-dimensional version (11D-ASC) of the 5-Dimensional Altered States of Consciousness Rating Scale [5D-ASC; 31], translated into Norwegian. The 11D-ASC questionnaire has 11 subscales: 1. *experience of unity*, 2. *spiritual experience*, 3. *blissful state*, 4. *insightfulness*, 5. *disembodiment*, 6. *impaired*

*cognition and control*, 7. *anxiety*, 8. *complex imagery*, 9. *elementary imagery*, 10. *audio-visual synaesthesia*, and 11. *changed meaning of percepts* [32]. For each statement in the questionnaire, participants were told to indicate their level of agreement on a visual analogue scale (VAS) anchored from "No, not more than usual" (left) to "Yes, much more than usual" (right), and scored by using percentage (left to right). An example of a statement is "I saw things I knew were not real". Participants were instructed to respond considering the time interval from when they felt they had an effect of the drug until the effects subsided, and where the "usual state" was before ketamine administration. Thus, the score indicated strength of experience relative to the normal non-psychedelic state. The mean score of all the questions gives the global-ASC score. The 11D-ASC was administered 45–60 minutes after discontinuation of ketamine, when noticeable drug effects had subsided.

## Set-up

Individual T1 weighted structural MR images (Phillips 3.0T Ingenia MR system, Philips Healthcare, The Netherlands) were obtained from each participant for spatial navigation to precisely locate the cortical target for TMS stimulation. For neuro-navigation we used the PowerMag View! system (MAG & More GmbH, München, Germany). This system uses two infrared cameras (Polaris Spectra) to track the position of the participant's head and TMS coil in space. A figure-eight coil (Double coil PMD70-pCool, MAG & More GmbH, München, Germany) was used for stimulation (maximum field strength of 2T (~ 210 V/m), pulse length of 100 μs, winding diameter of 70mm, biphasic pulse form) driven by a PowerMag Research 100 stimulator (MAG & More GmbH, München, Germany). The RMT was determined using PowerMAG Control (MAG & More GmbH, München, Germany). EEG was recorded with two 32-channel TMS-compatible amplifiers (BrainAmp DC, Brain Products, Germany) connected to a 60 channel TMS-compatible EEG cap. In addition, two electrodes detected eye movements (EOG), and a common reference was positioned at the forehead with the ground electrode. The impedance of all EEG electrodes was kept under 10 kΩ. EEG signals were sampled at 5000 Hz with 16-bit resolution and a 1000 Hz low pass filter was applied upon acquisition.

## Analysis

**TMS-EEG pre-processing and PCI analysis.** All pre-processing of the TMS-EEG data was done manually using the MATLAB (MATLAB R2016A, The Mathworks) based SiSyPhus Project (SSP 2.3e, University of Milan, Italy). EEG responses to TMS were visually inspected by a trained analyst to identify artefactual trials and channels containing abnormal amplitude activity which were excluded from further analysis. The interval around the time of the TMS-pulse (-2ms to 5ms) was removed for all participants to exclude the TMS artefact, and artefactual channels (flat, noisy or with high amplitude over a large duration of the recording) were interpolated. Trials with abnormal voltage traces (high variance, large transient deflections, movement artefacts etc.) in multiple channels were rejected and removed from further analysis. The remaining data were zero-centred (mean baseline correction; -500ms to -5ms) to eliminate any voltage offsets. Any residual artefacts in the data after 5ms was detected by independent component analysis (ICA) and then removed. Identification of artefactual components was done manually by inspecting EEG component topography, activity over time and power spectrum (S2 Fig). In order to minimize the risk of discarding brain activity signals, only components that were clearly dominated by eye blinks, eye movements and other muscle and non-physiological artefacts were removed [33, 34]. Components that were of unclear origin were kept in the data [35]. In general, we aimed to stay as close as possible to the analysis

described by the developers of PCI [12–14]. Thus, signals were referenced to the common average reference, using a 1 Hz high pass filter and 45 Hz low pass filter (Hamming window sinc FIR filter) to avoid the need for a notch filter to remove line noise, and downsampled to 312.5 Hz. The filter order was kept to its default value (3*rate/2).

The remaining analyses used for PCI calculation were fully automatic and performed by use of MATLAB scripts (MATLAB R2013A, The Mathworks) courtesy of Adenauer Casali (University of Milan, Italy) as described in [12]. First, source estimation of significant cortical sources was done by using a standard head model from the Montreal Neurological Institute (MNI) atlas. Then, the significantly active cortical sources were estimated using a threshold set by the 99th percentile of the distribution of maximum amplitudes of bootstrap resampled baseline activities before TMS pulse. If the amplitude of the source at a specific time exceeded this threshold, it was given the value 1, otherwise it was given the value 0. This resulted in a binarized matrix of significant sources over time, time-locked to the TMS pulse. Data in the interval 8–300 ms after the pulse of the resulting binarized matrix was used to derive the PCI value by calculating the Lempel-Ziv complexity (LZc), using the LZ76 compression algorithm. Finally, the LZc was normalized by the asymptotic maximum complexity of a matrix of the same size containing the same proportion of 1s to 0s, yielding the PCI value for the session [12, see supplemental material for a detailed explanation].

To avoid instability in the PCI values, a threshold for source entropy was set to 0.08, in accordance with [12]. All of the TMS-EEG recordings exceeded this threshold (*mean* = 0.6, *SD* = 0.2) and all the data showed a high signal-to-noise ratio (SNR >2, S3 Fig).

**Spontaneous EEG pre-processing and signal diversity analysis.** The spontaneous EEG data were pre-processed using EEGLAB. First, the data were split into 8-second non-overlapping segments, resulting in 15 epochs per condition (normal wakefulness and during ketamine infusion, with eyes open and eyes closed). Artefactual channels were marked, removed, and interpolated. Epochs were rejected based on the same criteria as for the TMS-EEG data and the data were baseline corrected (zero mean for the full length of the epoch). Signals were referenced to the common average reference, filtered with a 0.5 Hz high pass and 45 Hz low pass filter (Hamming window sinc FIR filter, default filter order value (3*rate/2)), and downsampled to 250 Hz. ICA components such as eye blinks and eye movement were manually detected and excised from the data. As in [10], the surface Laplacian of the data was computed, increasing topographical specificity by subtracting the averaged signal of each channels' nearest neighbours [36]. From the 62 channels, only 9 were chosen for signal diversity analysis (S3 Fig) due to the entropy measures being calculated based on the distribution of states observed in the data. Since number of states, S, available in a network of N binary nodes is $S = 2^N$, and the measures require an estimate of the probability density distribution over states, the number of samples in an epoch should be at least as large as the number of states available to yield a representative estimate of the underlying distribution of states. With each epoch containing 2000 samples (8s epoch length x 250Hz sampling rate), the maximal number of channels that could be included in the analysis if all states were to have a chance of being sampled at least once, would be 10 (# available states = 2^10 = 1024). However, using a simulation study we estimated the number of samples required to expect sampling each state at least once to be ~2100 even with only 9 channels included (results not shown). Thus, to increase the chance of getting a decent sampling of the distribution, even in a condition yielding the maximum entropy distribution of coalition, we decided to use only 9 channels. It should be noted that our estimation assumes that the samples are independent, which is not a good assumption for EEG data, indicating that our inclusion of 9 channels may, if anything, be too optimistic with regards to obtaining a good estimate of the distribution of coalitions.

The signal diversity measures amplitude coalition entropy (ACE), synchrony coalition entropy (SCE), and Lempel Ziv Complexity (LZc), were calculated as described in [10]. We first performed a Hilbert transformation and then binarized the data. The binarization threshold was set to the mean absolute amplitude (ACE, LZc) or to the absolute phase synchrony between each channel pair according to a 0.8 radian threshold (SCE). Next, we found the distribution of states over time, defining a state as a binary string of the activity over channels (ACE) or phase synchrony over channel pairs (SCE) at a given time point. Shannon entropy was then calculated over the state distributions and normalized according to the maximum entropy of a randomized sequence with similar characteristics as the original (ACE, SCE). Finally, the mean values were calculated over epochs (ACE) or channel pairs and epochs (SCE). LZc was calculated by directly applying the LZ76 algorithm [37, 38] to the spatially concatenated binarized activity matrix. Normalization was done by dividing the resulting raw value with the LZc of the same data shuffled in time.

**Statistical analysis.** All statistical analyses were done in SPSS (IBM SPSS Statistics 24). Normality was determined by the Shapiro-Wilk test (significance $p > 0.05$). To investigate differences between the sub-anaesthetic ketamine condition and the normal wakeful state on PCI and RMT we used the parametric paired-samples T-test and the non-parametric Wilcoxon signed-rank (WSR) test, respectively. A linear mixed model was used to assess whether changes in stimulation intensity affected the spatiotemporal activation values (average of the significant cortical sources activated after TMS) and PCI values. A random intercept was included to account for within-subject correlations. The model parameters were estimated using restricted maximum likelihood, and statistical significance was assessed by t-tests. To assess the effects of sub-anaesthetic ketamine, relative to normal wakefulness, in eyes open compared to eyes closed condition, we used a repeated two-way ANOVA with spontaneous signal diversity measures as dependent variables (LZc, ACE, SCE). The analysis was designed to uncover effects of the conditions (normal vs ketamine and eyes open vs eyes closed) as well as any interaction effect between the conditions. To assess the relations between PCI values and stimulation intensity, PCI and spontaneous signal diversity measures and continuous and total ketamine dose, and total ketamine dose and global-ASC score, we used Spearman's rank order correlation. Moreover, PCI and spontaneous signal diversity in the ketamine condition was correlated with the 11-ASC and global-ASC. Significance threshold for all tests was $p < 0.05$.

## Results

### Increased spontaneous, but not evoked signal diversity in sub-anaesthetic ketamine compared to normal wakefulness

Our main aim was to investigate whether information integration and differentiation as measured by PCI, and spontaneous signal diversity as measured by LZc, ACE, and SCE, is affected by sub-anaesthetic doses of ketamine compared to normal wakefulness. For PCI, we first investigated the spatiotemporal response to TMS (Fig 1) and found that both in the normal wakeful state and the sub-anaesthetic condition, the brain responded to TMS with long-lasting patterns of activation. The response was not limited to the site of activation, as seen by the distribution of significant sources, but spread to different cortical locations. Qualitatively, the spatiotemporal characteristics appeared similar in both conditions, with small variations in amplitude and latencies between the two conditions for individual participants. In accordance with these observations, we found no significant difference between PCI values for normal wakefulness (*mean* = 0.53, *SE* = 0.02) and sub-anaesthetic ketamine (*mean* = 0.55, *SE* = 0.03, *t* (9) = -0.87, *p* = 0.41, *r* = 0.27, Fig 2).

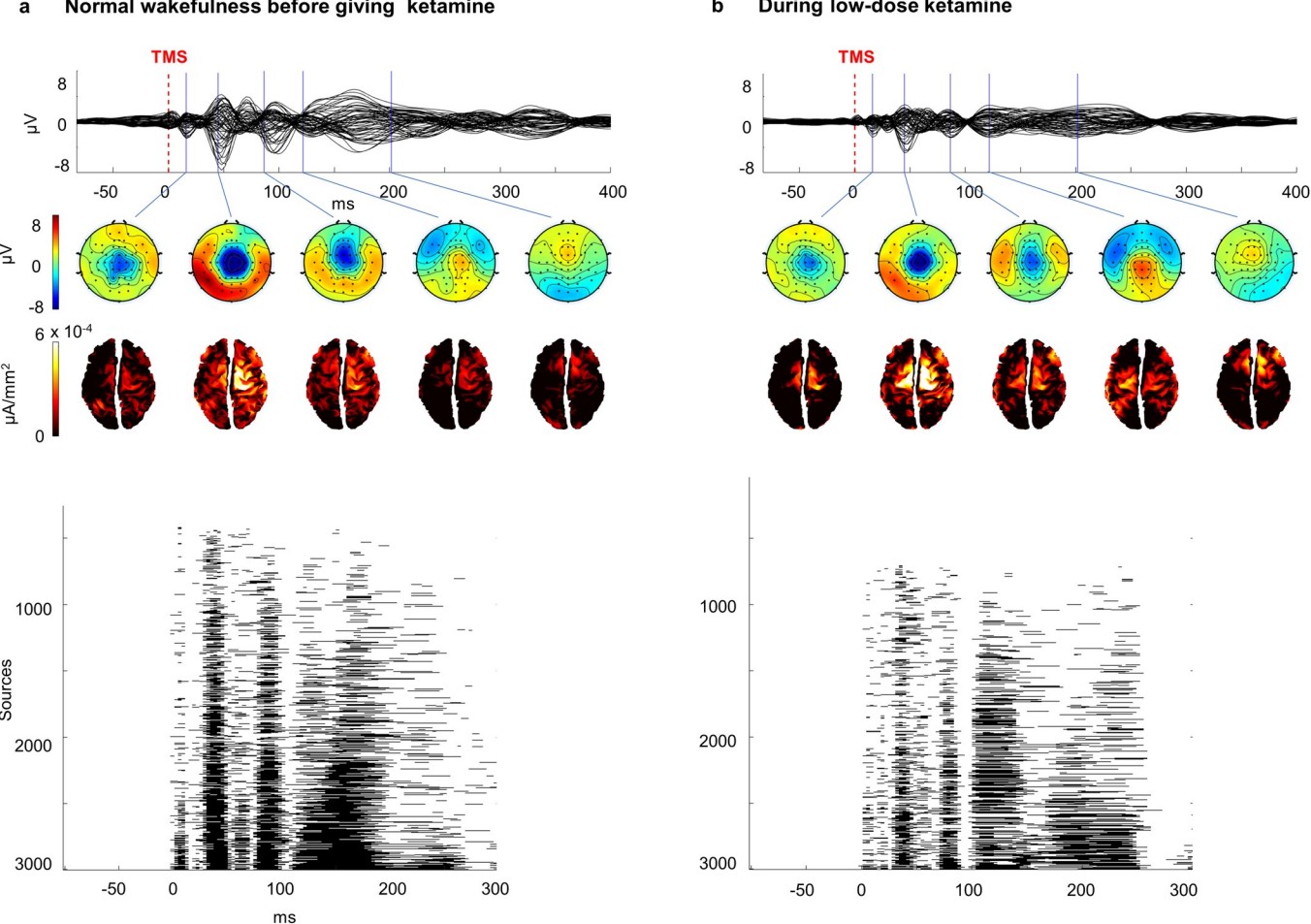

**Fig 1. Spatiotemporal dynamics of TMS-EEG responses in normal wakefulness versus ketamine-induced psychedelic state.** Averaged potentials evoked by local transcranial magnetic stimulation (TMS) over all EEG channels (298 and 281 trials) in one representative subject (a) before and (b) during ketamine administration (stimulation location was right BA 7). The TMS stimulation intensities were 80% and 79% of the maximal stimulator output, respectively. Upper panels: TMS-evoked potentials for all channels. Middle panels: voltage topographies, of selected latencies, reflecting the electrical activity across the scalp and corresponding distributions of significant cortical currents. Bottom panels: binary SS(x,t)-matrices where significant sources at a given time are displayed as black. The sources are ordered from bottom to top according to total amount of significant activation in the response after TMS.

For the spontaneous signal diversity measures (LZc, ACE and SCE), there was no statistically significant two-way interaction between condition (normal wakefulness and sub-anaesthetic ketamine) and participants having eyes open or closed (LZc: $F(1,9) = 0.85$, $p = 0.38$, ACE: $F(1,9) = 1.44$, $p = 0.26$, and SCE: $F(1,9) = 1.27$, $p = 0.29$). There was, however, a significant main effect of ketamine (normal wakefulness vs. sub-anaesthetic ketamine infusion) (LZc: $F(1,9) = 11.13$, $p < 0.05$, $r = 0.75$, ACE: $F(1,9) = 10.67$, $p < 0.05$, $r = 0.74$ and SCE: $F(1,9) = 11.79$, $p < 0.05$, $r = 0.75$), and a significant main effect of having eyes open compared to closed (LZc: $F(1,9) = 20.83$, $p < 0.05$, $r = 0.84$, ACE: $F(1,9) = 17.78$, $p < 0.05$, $r = 0.81$, and SCE: $F(1,9) = 16.71$, $p < 0.05$, $r = 0.81$). Post-hoc analyses with a Bonferroni adjustment revealed that all measures significantly increased from wakefulness to the sub-anaesthetic ketamine condition (LZc: $0.01 \pm 0.003$, p < 0.05, ACE: $0.02 \pm 0.005$, p < 0.05, SCE: $0.01 \pm 0.003$, p < 0.05), and decreased when participants had eyes closed compared to eyes open, across drug condition (LZc: $-0.02 \pm 0.004$, p < 0.05, ACE: $-0.03 \pm 0.01$. p < 0.05, SCE: $-0.02 \pm 0.04$, P < 0.05).

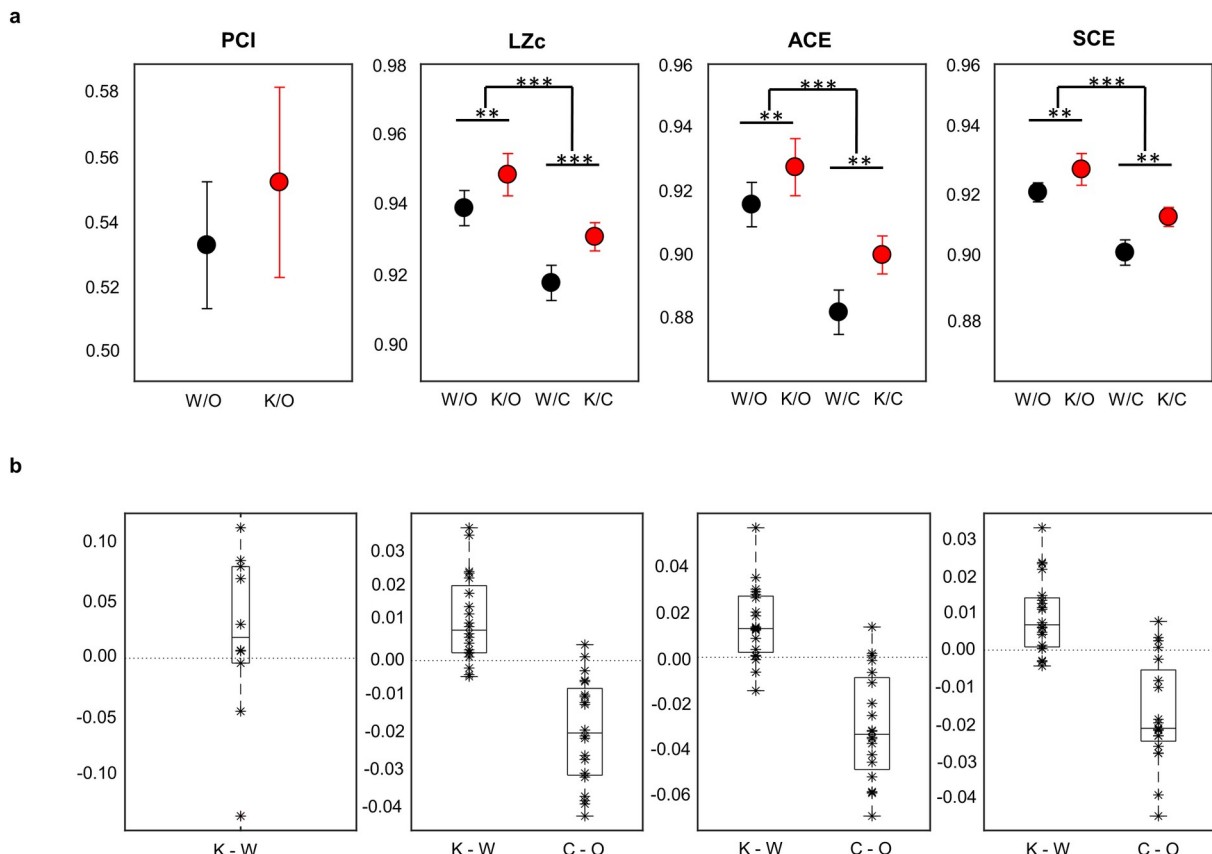

**Fig 2. Average values and difference values of the Perturbational Complexity Index (PCI), and three measures of spontaneous EEG signal diversity (LZc, ACE, and SCE). a)** Average values with one standard error of the mean (SEM) error bars for PCI, LZc, ACE, and SCE in wakefulness with eyes open (W/O) and eyes closed (W/C), and ketamine eyes open (K/O) and eyes closed (K/C). The stars (*, **, and ***) indicate statistical significance (p <0.05, p <0.01, p < 0.001) between wakefulness and ketamine, and eyes open or closed. **b)** Boxplots showing differences in individual PCI, LZc, ACE, and SCE values subtracting ketamine from wakefulness (K-W) and eyes closed from eyes open (C-O).

## Signal diversity values were unaffected by stimulation intensity and sub-anaesthetic ketamine dose

Since RMT was measured before and after sub-anaesthetic ketamine to determine the intensity for TMS stimulation, we wanted to investigate whether stimulation intensity affected the PCI values. First, we found no significant difference in RMT before (*median*: 54.5, *range*: 46.5–64) and after ketamine administration (*median*: 56.8, *range*: 43–62, $z$ = -0.48, $p$ = 0.64, $r$ = -0.11). Moreover, the mean change in stimulation intensity relative to maximum stimulator output from before to after sub-anaesthetic ketamine was a 0.81% decrease (*SD*: -9.7%– 14.6%), but this change in stimulation intensity did not significantly affect spatiotemporal activation values (*regression coefficient* [95% CI] = $-3\times10^{-3}$, [$-4\times10^{-3}$ $12\times10^{-2}$], $p$ = 0.35), nor PCI values (*regression coefficient* [95% CI] = -0.04, [$-8\times10^{-3}$ 0], $p$ = 0.07). Intra-class correlation coefficients were found to be 0.88 for spatiotemporal activation and 0.31 for PCI.

For ketamine doses, we found no significant correlation between the rate of continuous maintained ketamine dose and PCI ($r_s$ = -0.04, $p$ = 0.90) or spontaneous signal diversity (LZc eyes open: $r_s$ = 0.04, $p$ = 0.92, LZc eyes closed: $r_s$ = 0.05, $p$ = 0.89, ACE eyes open: $r_s$ = 0.05, $p$ = 0.89, ACE eyes closed: $r_s$ = -0.04, $p$ = 0.92, SCE eyes open: $r_s$ = 0.03, $p$ = 0.93, SCE eyes closed: $r_s$ = 0.20, $p$ = 0.58). Similarly, we found no significant correlation between total ketamine doses and PCI values ($r_s$ = 0.18, $p$ = 0.63) or spontaneous signal diversity values (LZc

eyes open: $r_s$ = -0.03, $p$ = 0.93, LZc eyes closed: $r_s$ = 0.15, $p$ = 0.68, ACE eyes open: $r_s$ = 0.02, $p$ = 0.96, ACE eyes closed: $r_s$ = 0.10, $p$ = 0.78, SCE eyes open: $r_s$ = 0.07, $p$ = 0.86, SCE eyes closed: $r_s$ = 0.26, $p$ = 0.47).

## Correlations with alterations in phenomenology

All participants retrospectively reported to have had an effect of ketamine, but to differing degrees (Fig 3A). The overall average response to the 11D-ASC showed that the subscales *disembodiment*, *complex imagery*, and *elementary imagery* had the highest scores, while the subscale *anxiety* had the lowest score (Fig 3B). No significant correlations were found between total ketamine dose and *global-ASC* scores ($r_s$ = 0.33, $p$ = 0.35). The subscales *experience of unity* and *anxiety* had high correlations with the signal diversity measures in the eyes open condition (*experience of unity* and ACE: $r$ = -0.52, $p$ = 0.13, *anxiety* and PCI: r = 0.50, p = 0.14, *anxiety* and LZc: r = 0.72, $p$ < 0.05), while *complex imagery* and *elementary imagery* had the highest correlation in the eyes closed condition (*complex imagery* and LZc: r = -0.59, $p$ = 0.07, *elementary imagery* and LZc: r = -0.53, $p$ = 0.12, Fig 3C).

## Discussion

The main result of this study is the observation of significantly increased values of spontaneous EEG signal diversity measures in the ketamine-induced psychedelic state induced by sub-

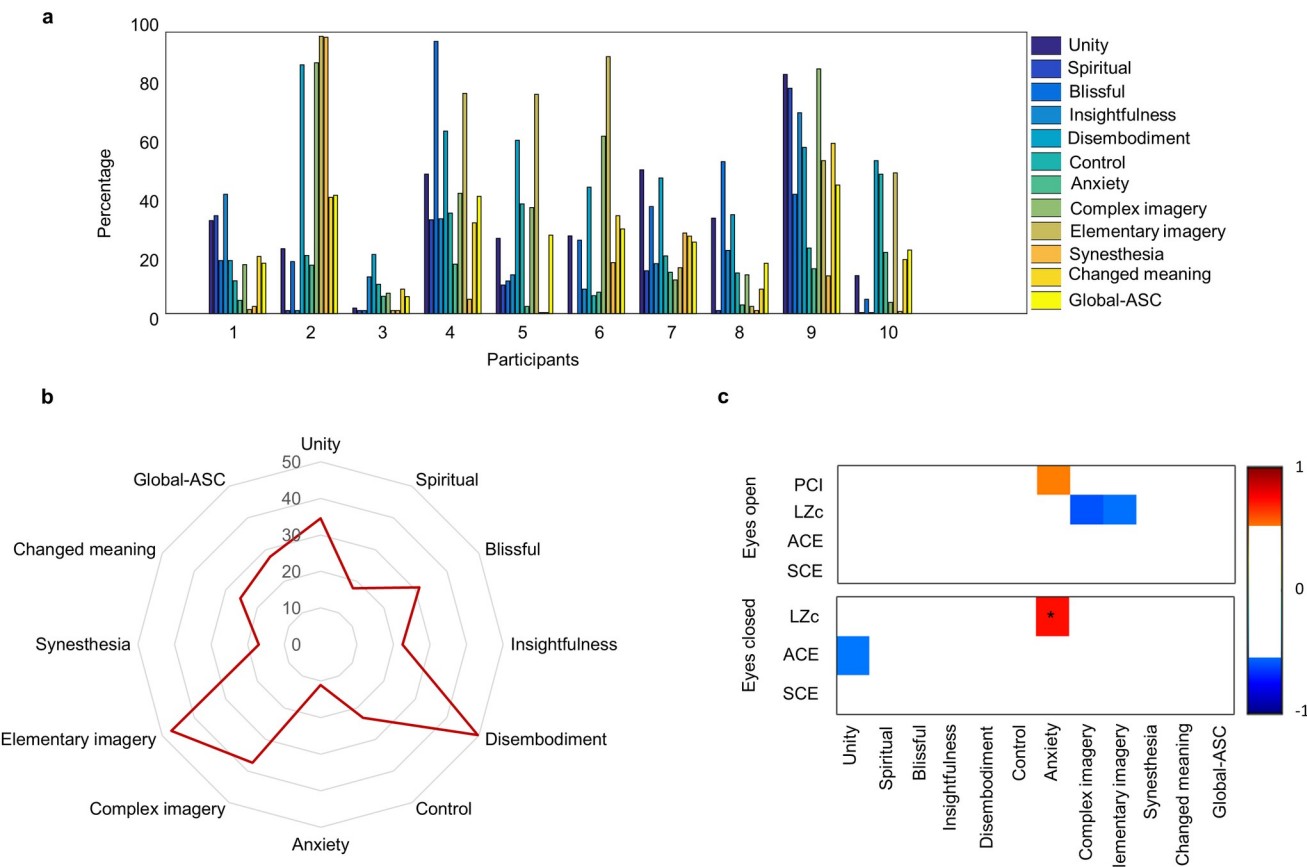

**Fig 3. Phenomenology of the psychedelic state induced by sub-anaesthetic ketamine. a**) Individual scores of the 11-Dimensional Altered States of Consciousness Rating Scale (11D-ASC) questionnaire, Global-ASC indicates the average score over all dimensions, **b**) Total mean scores for each dimension of the ASC questionnaire, **c**) Correlation across signal diversity measures (difference scores between sub-anaesthetic ketamine and wakefulness) and ASC scores. Weak correlations (-0.5 < r < 0.5) are omitted (white) to only highlight strong correlations. Significance is indicated with a star (*).

anaesthetic doses of ketamine, compared to the normal wakeful state. Moreover, we found a significant difference in spontaneous signal diversity in the eyes closed condition compared to eyes open during both ketamine administration and normal wakefulness. In contrast, we observed no significant difference in PCI between the normal wakeful state and the ketamine-induced psychedelic state. As both PCI and spontaneous signal diversity have been seen to vary with the level of wakefulness in humans [10, 12, 16, 17], these results suggest that PCI and spontaneous signal diversity measures may be sensitive to different aspects of conscious states.

## Why did spontaneous signal diversity, but not PCI values, increase in the psychedelic state?

In both the normal wakeful state and sub-anaesthetic ketamine state, the spatiotemporal EEG responses to TMS perturbations seemed to contain fast, high-amplitude, and long-lasting waves of activity (Fig 1) resulting in no significant differences in PCI values (Fig 2). These results are consistent with the findings of [13] where the PCI values measured during ketamine anaesthesia (i.e. with higher doses causing a state of unresponsiveness) were similar to those measured during normal wakefulness. In contrast, the spontaneous signal diversity measures (LZ, ACE and SCE) significantly increased during sub-anaesthetic ketamine compared to the normal wakefulness, in accordance with previous findings [17, 22]. The difference between PCI and other signal diversity measures may seem unexpected, since previous studies have indicated that measures of spontaneous electrophysiological signal diversity (e.g. LZc, ACE, and SCE) correspond quite well with PCI across conditions such as sleep [10] and propofol anaesthesia [16]. Why, then, did we find that sub-anaesthetic ketamine increased spontaneous signal diversity values but not PCI values?

One possible explanation for the observed differences could be that for PCI, complexity is computed from evoked EEG responses, unlike LZc, ACE and SCE which are computed from spontaneous EEG. While evoked and spontaneous signal diversity measures reflect differentiation (diversity of patterns over time) [10], TMS evokes EEG responses that reflect causal interactions in the brain and the spatiotemporal spread of which indicates degree of integration [12, 20]. In contrast, signal diversity or complexity in spontaneous EEG is not necessarily strongly related to causal interactions but could for example depend on degrees of independent vs. common driver inputs, thus being related to functional connectivity measures [20, 39]. As such, PCI is considered to more closely reflect concurrent integration and differentiation by assessing the so-called "deterministic" responses to TMS, i.e. the responses that remain after averaging multiple trials [12]. Crucially, PCI is designed to probe the general capacity of the brain to engage in complex causal interactions while being insensitive to the specific pattern of ongoing neural activity [3]. This might explain the discrepancy between the PCI and spontaneous signal diversity results obtained here and by [17].

One aspect that strongly distinguishes signal diversity measures from PCI is that the signal diversity measures will increase when processes in the brain grow increasingly noisy or complex, while PCI will only increase if the underlying brain processes get more complex. This is largely an issue related to the fact that signal diversity measures are dependent on observational data, while the PCI is calculated from controlled perturbations and subsequent observations of the brain. Thus, conditions where signal diversity measures increase, but PCI remains unchanged may be an indicator that the brain state is more chaotic, but not more complex.

By design, spontaneous signal diversity measures are more sensitive than PCI to changes related to the complexity of the ongoing brain network activity. Our observations of the effects of eye closing seem to support this idea. We found that for all spontaneous signal diversity measures, there was a significant decrease when participants had their eyes closed compared

to when they had open eyes. This was independent of whether the participants were in a normal wakeful state or in the sub-anaesthetic ketamine state. Spectral analyses show that power spectrum density differences between wake and ketamine change for spontaneous data with eyes open and eyes closed, as well as for perturbational data, with different frequency band changes for each condition (S4 Fig). This is in line with findings of a lack of visual input increases synchronous alpha band activity [40], thus ultimately affecting signal diversity. However, because drastically simplifying visual stimuli by closing the eyes might significantly reduce the complexity of the conscious content by reducing an aspect of the participant's phenomenological experience, these findings suggest that spontaneous signal diversity may be more related to conscious content rather than conscious state. In contrast, having eyes closed does not seem to affect the TMS-EEG response [41, supplemental material] nor the PCI values during wakefulness [12, supplemental material]. Although perturbational data show frequency band changes in wake versus ketamine data, PCI seems less sensitive to changes in content (e.g. induced by removing visual aspects from phenomenal experience) within a global physiological state (here: being fully awake).

Moreover, the spectral changes in spontaneous data caused by sub-anaesthetic doses of ketamine may also be expected to influence signal diversity values. For example, sub-anaesthetic doses of ketamine have been shown to decrease alpha power in parallel with subjective ratings of dissociation of experience [42] as well as increase gamma power [43, 44]. However, using phase-shuffling normalization, Schartner et al. [17] found that the spectral profile changes seen in sub-anaesthetic ketamine could not explain the increased spontaneous signal diversity observed in the psychedelic condition.

Furthermore, the lack of significant change in PCI values in the psychedelic state compared to normal wakefulness (Figs 1 and 2) can be interpreted as low doses of ketamine causing so small changes in differentiation and integration that no change is detected in the TMS-evoked responses. However, functional MRI studies have shown that hallucinogenic drugs produce an increased repertoire of activity patterns, thus increasing neural entropy compared to normal wakefulness [45, 46] and reflecting increased differentiation of brain activity as observed here and previously [17, 42]. An alternative hypothesis is that the increased signal diversity indicating neural differentiation may occur in parallel with a reduction of integration within the relevant brain networks, thus leading to a zero net change in PCI value. For example, sub-anaesthetic doses of ketamine reduce fronto-parietal effective connectivity compared to normal wakefulness [44], and a similar decrease has been found in anaesthetic doses of ketamine [47]. Yet, given the blocking effect of ketamine on NMDA-receptors and the widespread importance of these receptors throughout the cortex, it seems unlikely that reduced effective connectivity is limited to frontoparietal networks and it remains to be determined exactly how this decrease is related to overall cortical functional integration.

## Relating signal diversity to phenomenological changes

The increase of signal diversity in the sub-anaesthetic ketamine condition may result from the brain state changes that cause the psychedelic experiences. Although all participants reported having had an effect of ketamine, the degree of subjective psychedelic effect, as measured by 11D-ASC, varied between participants (Fig 3A), which could be due to subjective differences in the quantification of the effect, different individual reactions to ketamine, or due to differences in administered ketamine dose.

Similar to previous findings from placebo-controlled studies with sub-anaesthetic ketamine [32, 48], the subscales *disembodiment* (feeling of dissociation between mind and body) and *elementary imagery* (changes in visual imagery with eyes closed) had the highest overall 11D-ASC

scores, while anxiety had the lowest overall scores (Fig 3B). Thus, the drug effects were largely as expected. The subscales *experience of unity and anxiety* had highest correlations with the signal diversity measures in the eyes open condition, while complex imagery and elementary imagery had the highest correlation with eyes closed (Fig 3C). However, only the correlation between the anxiety subscale and signal diversity was significant. In comparison, others [17] have found that increased spontaneous signal diversity in sub-anaesthetic ketamine was correlated with *overall intensity* of psychedelic experience as well as *ego-dissolution* and *vivid imagination, which corresponds to the* subscales *disembodiment* and *complex imagery* of the 11D-ASC. It is not clear why our results differ with previous studies, and why anxiety would have a positive, significant correlation with signal diversity, although there is some evidence that ketamine may be useful for treatment of depression and anxiety disorders [49]. Further investigations are probably needed to further test and clarify the reasons for these apparent correlations.

## PCI and spontaneous signal diversity may reflect complementary aspects of consciousness

Behaviourally unresponsive states, such as the unresponsive wakefulness ("vegetative") state, where subjects lack behaviourally verifiable intrinsic experiences, are often associated with low PCI values compared to normal wakefulness [12, 14]. However, states in which subjects are behaviourally unresponsive but give delayed reports of having had vivid conscious experiences such as dreams during anaesthetic ketamine [13] and dreams during REM sleep [12], are associated with high PCI values similar to normal wakefulness. Therefore, PCI may reflect the brain's capacity for sustaining experience per se, without differentiating whether the experience occurs with or without extrinsic awareness or ability to respond [12]. Furthermore, since we did not find any difference in PCI comparing the sub-anaesthetic ketamine condition with normal wakefulness (all values in the ketamine condition were within the range of values typically reported for individuals in the awake state [14], PCI may be more useful for differentiating between brain states with or without the capacity for sustaining consciousness, rather than measuring gradations of conscious content in wakeful experience.

Given that PCI values beyond the range found in normal wakefulness (i.e. up to 0.7) have so far not been measured [14], this poses the question of whether it is conceivable that there may exist brain states that give a measurably higher PCI value compared to the normal awake state. The psychedelic state, which has been described as involving "unconstrained cognition" [50], has been considered a possible candidate. However, such changes in cognition may not be sufficient to exert a net change (increase) in PCI, suggesting some sort of "enhanced consciousness". Rather, the psychedelic state might be a more "expansive" state in terms of "flexibility" in cognition compared to normal wakefulness, which could be reflected by increased signal diversity or entropy in the brain [17, 45, 46]. Spontaneous signal diversity measures might therefore reflect increases in the complexity of conscious content, although one would expect significant correlations with phenomenology, which were only found for anxiety in the present study. Even though conscious content were to be modulated in the psychedelic state, this may not necessarily mean that the level of consciousness [i.e. level of arousal or wakefulness, 51] is altered. This also holds for having eyes open or closed, where PCI does not change, but signal diversity does. Furthermore, if LZc, ACE, and SCE only reflects the complexity of conscious content, and this complexity of content is separable from the capacity for consciousness, it makes sense that these measures correlate with PCI when comparing wakefulness with sleep [16] or anaesthesia [10]. This is because the conscious content normally appears to correlate with the level of wakefulness in these conditions [51].

## Limitations

A limitation of our study is the moderate sample size. 34 participants were included in the study, but only 10 completed. Because of strict inclusion criteria, for example finding the RMT and obtaining high TEPs, there is a possibility that the selection of subjects might have biased the results. Moreover, after 10 participants, we performed an interim analysis of the TMS-EEG data to evaluate continuation of the study and inclusion of more participants. As we did not find any significant differences in PCI values, and power tests could not predict significant differences nor correlations between PCI values and the 11D-ASC questionnaire with increased sample size, we decided not to continue. A small sample size is associated with higher variability and lower reliability of statistical analyses, for example of correlation, also limiting the investigation of interaction effects of variables such as gender and age. Future studies might thus benefit from a larger sample size. However, the sample size used in the current study is larger than in [13] where no difference in PCI was found for anaesthetic doses of ketamine compared to normal wakefulness, and our spontaneous signal diversity results and correlation finds are similar to [17] where number of participants was higher.

Using the RMT to determine intensity of stimulation has been debated due to distance of coil to cortex differing between cortical areas [52, 53] and differences in anatomy between motor and non-motor cortical stimulation sites [54]. However, since racemic ketamine has shown to affect cortical excitability [55], not adjusting the stimulation intensity to the RMT could lead to PCI values reflecting differences in excitability and not the psychedelic experience per se. Moreover, RMT values did not significantly differ before and after ketamine administration and change in stimulation intensity from wake to ketamine condition had a mean of 0.8% decrease, most likely not affecting the TEPs [12, 56]. Finally, we did not find a significant effect of stimulation intensity on spatiotemporal activation, nor on PCI values.

To mitigate influences of auditory evoked potentials on PCI values, participants listened to noise masking sounds based on the TMS clicks. However, we did not use foam padding to minimize scalp sensations, nor were participants subject to a sham rTMS session, or given a questionnaire to completely exclude effects of audition.

The order of the experiments was fixed which could have influenced the results of the study, in addition, we administered ketamine to the participants as a continuous infusion, gradually increasing the dose until the participants reported an effect of the drug instead of giving a predefined bolus and maintenance dose. The goal was to ensure that all participants had comparable drug effects on subjective experiences. However, as the infusion was not always increased in equal steps (S5 Fig) there were different continuous infusion rates and total doses of ketamine for each participant (particularly for the first 3 participants, where the doses were increased in smaller steps and over longer time than for the other 7 participants. A bolus dose of ketamine [43, 57] could have avoided this complication, and allowed for placebo control, but would not have ensured comparable subjective drug effects for all participants. Although the differences in infusion rate and dose may have affected PCI and 11D-ASC scores, we found no significant correlation between total and continuous dosage (infusion rate or total dose), nor between total dose and subjective experience. Although, we cannot exclude the possibility that expectation bias affected the participants' evaluation, our study confirmed findings from similar studies that included placebo control [17]. Placebo control is anyway difficult for psychedelic states when it is particularly easy for a participant to determine whether they got an active or inactive compound.

## Future directions

Further exploring how different proposed markers of consciousness are affected in conditions where phenomenal content is altered may help us understand which aspects of consciousness

the markers are most closely linked to. Understanding how these markers respond to conditions that induce changes in phenomenology, by pharmacology (e.g. anaesthetics or psychedelics), pathology (e.g. in patients with psychiatric disorders or disorders of consciousness), or physiological brain states (e.g. sleep, in "flow states", or sensory deprivation) can help us understand the relations between functional, dynamical, or structural properties of the brain and conscious experience. Thus, further testing, comparing, and contrasting promising neurophysiological markers of consciousness in conditions where phenomenology is altered and mapping out the relation between observed changes in the markers and reported changes in the structure of experience, may help identifying sets of complementary markers that are sensitive to distinct aspects of conscious experience. Furthermore, it may help deciding whether unnecessarily complex measures can be exchanged for a set of simpler markers for some practical purposes.

## Conclusion

In this study we investigated whether TMS-evoked (PCI) and spontaneous EEG signal diversity measures (LZc, ACE, and SCE) are affected by sub-anaesthetic doses of ketamine causing a psychedelic state, compared to the normal wakeful state. We found no significant change in the perturbational complexity index (PCI) when comparing sub-anaesthetic ketamine and normal wakefulness, but we did find that spontaneous EEG signal diversity was significantly higher when participants were under the influence of sub-anaesthetic doses of ketamine than in normal wakefulness. We also found that the spontaneous signal diversity measures were significantly lower with eyes closed than with eyes open, both during normal wakefulness and under the effect of sub-anaesthetic ketamine doses. Furthermore, we found correlations between changes in spontaneous signal diversity measures and subjective ratings of phenomenological experience. These results suggest that spontaneous and evoked measures of EEG signal diversity may reflect distinct, complementary aspects of changes in brain function related to altered states of consciousness. Spontaneous EEG measures may thus capture properties related to the content of consciousness, while evoked measures (PCI) may index the system's capacity for consciousness.

## Supporting information

**S1 Fig. Flowchart of participation in the different stages of the study.** The number of participants included (left) or discontinued (right) at each stage of the study is indicated with *n*. In the end 10 participants completed the study.
(TIF)

**S2 Fig. Example of artefactual components.** Example of (a) eyeblink artefact and (b) muscle artefact. The left panel shows component activity for all trials (above) and the average of all trials (below) and to the right. The right panel shows power spectrum and scalp map power for the components.
(TIF)

**S3 Fig. Electrode positions, signal-to-noise ratio (SNR), and source entropy (H).** (a) Spontaneous signal diversity was calculated from 9 channels (yellow) distributed across the scalp: F5, Fz, F6, C6, Cz, C6, P5, Pz, and P6. (b) Signal-to-noise ratio (SNR) and source entropy (H) for all TMS-EEG measurements and participants. The black dots are participants during normal wakefulness and the red dots are participants during sub-anaesthetic ketamine. The source entropy of the data was above 0.08 and all data showed a high SNR ($> 2$). None of the data

were excluded based on too low source entropy.
(TIF)

**S4 Fig. Spectral characteristics of the data.** Here we show examples of processed data from within a single epoch in the awake (a) and ketamine (b) conditions, as well as mean power spectral densities (PSD) across participants for wakefulness and ketamine (c) and the within subject, pairwise mean difference between wake and ketamine PSDs (d). Each of these are shown for the recordings of spontaneous activity in the resting state with eyes open (column 1) and resting state with eyes closed (column 2), as well as in response to TMS stimulation with eyes open (third column). In (a) and (b), the thick line shows the activity of channel Cz in the single epoch, while the gray lines in the background shows the activity of the remaining channels in that same epoch. For column 3, the black dashed line shows the time of the TMS pulse. In (c), the PSD is shown between 0 and 45 Hz, for the wakefulness (black) and ketamine (red) conditions. The thick line indicates the mean across participants, while the shaded area shows the extent of 3 standard errors of the mean. For the third column, the PSD is calculated only for the data after the pulse (0-500ms). The thick blue line in (d) shows the mean pairwise difference in the PSD (within participants) while the shaded area again shows the extent of 3 standard errors of the mean. Here we see that the frequency bands changes in the ketamine versus wakefulness states differ between the data types: in the eyes open condition, differences appear in the beta and low gamma bands (~12-30Hz); in the eyes closed condition, differences appear in theta, alpha, and low gamma bands (~4–12 Hz and ~20–30 Hz); in the perturbed data, in the alpha and beta bands (~8–20 Hz).
(TIF)

**S5 Fig. Ketamine administration.** The plots show the time course of ketamine infusion rates, between 0.1 and 1.2 mg/kg/hr for the ten subjects. The infusion was eventually stabilized (0.1–1.0 mg/kg/hr) at a constant rate in the final period, during which, all the TMS-EEG data used in the study were collected. The total amount of ketamine given is shown to the right.
(TIF)

**S1 Table. Stimulation parameters, ketamine doses and PCI values for the TMS-EEG sessions.** PCI values in bold were used in the present study. R = right, L = left, BA = Brodmann Area, **%** = percentage of maximal stimulator output.
(TIF)

## Acknowledgments

We are thankful to Marcello Massimini, Silvia Casarotto, and Matteo Fecchio for providing computer software and insightful, technical help, and support regarding the PCI measurements, and to M. Massimini also for comments on the manuscript. We also thank Pål G. Larsson for technical advice regarding EEG, Franz Vollenveider for advice regarding ketamine administration, Brita Noorland for help with ketamine administration, and Anikó Kuztor for helping with the data acquisition and analysis. Co-author Bjørn E. Juel is presently also affiliated with The Department of Psychiatry, Center for Sleep and Consciousness, University of Wisconsin, Madison, WI, USA. Results from this work were previously presented at two conferences in 2017 [58, 59] and as a preprint manuscript in January 2019 [60].

## Author Contributions

**Conceptualization:** Nadine Farnes, Bjørn E. Juel, André S. Nilsen, Luis G. Romundstad, Johan F. Storm.

**Data curation:** Nadine Farnes, Bjørn E. Juel, André S. Nilsen, Luis G. Romundstad.

**Formal analysis:** Nadine Farnes, Bjørn E. Juel, André S. Nilsen.

**Funding acquisition:** Johan F. Storm.

**Investigation:** Nadine Farnes, Bjørn E. Juel, André S. Nilsen, Luis G. Romundstad.

**Methodology:** Nadine Farnes, Bjørn E. Juel, André S. Nilsen, Luis G. Romundstad, Johan F. Storm.

**Project administration:** Johan F. Storm.

**Resources:** Johan F. Storm.

**Supervision:** Bjørn E. Juel, André S. Nilsen, Luis G. Romundstad, Johan F. Storm.

**Visualization:** Nadine Farnes, Bjørn E. Juel.

**Writing – original draft:** Nadine Farnes.

**Writing – review & editing:** Nadine Farnes, Bjørn E. Juel, André S. Nilsen, Luis G. Romundstad, Johan F. Storm.

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
