## [Decision Letter · Decision Letter 0]

28 Aug 2020

PONE-D-20-19579

Increased signal diversity/complexity of spontaneous EEG, but not evoked EEG responses, in ketamine-induced psychedelic state in humans

PLOS ONE

Dear Dr. Farnes,

Thank you for submitting your manuscript to PLOS ONE. After careful consideration, we feel that it has merit but does not fully meet PLOS ONE’s publication criteria as it currently stands. Therefore, we invite you to submit a revised version of the manuscript that addresses the points raised during the review process.

We look forward to receiving your revised manuscript.

Kind regards,

Vladyslav Vyazovskiy, PhD

Academic Editor

PLOS ONE

Journal Requirements:

2. Thank you for including your ethics statement:  "This study was approved by the regional committees for medical and health research ethics (2015/1520/REK sør-øst A).".   

3.

We note that you have indicated that data from this study are available upon request. PLOS only allows data to be available upon request if there are legal or ethical restrictions on sharing data publicly. For more information on unacceptable data access restrictions, please see http://journals.plos.org/plosone/s/data-availability#loc-unacceptable-data-access-restrictions.

Reviewers' comments:

Reviewer's Responses to Questions

**Comments to the Author**

1. Is the manuscript technically sound, and do the data support the conclusions?

Reviewer #1: Partly

Reviewer #2: Yes

2. Has the statistical analysis been performed appropriately and rigorously? 

Reviewer #1: Yes

Reviewer #2: Yes

3. Have the authors made all data underlying the findings in their manuscript fully available?

Reviewer #1: Yes

Reviewer #2: Yes

4. Is the manuscript presented in an intelligible fashion and written in standard English?

Reviewer #1: Yes

Reviewer #2: Yes

5. Review Comments to the Author

Reviewer #1: The authors present a study showing changes in spontaneous electroencephalogram (EEG) signals - increased signal diversity/complexity - but not in evoked EEG responses in ketamine-induced psychedelic state in humans. The work is novel and the experiments seem to have been conducted rigorously. However, there are some issues with the manuscript that need to be addressed, and things that need to be clarified and explained in greater detail.

Major points:

- The authors mention several times the term "signal diversity" but fail to describe it from a mathematical point of view. What makes a signal diverse? Furthermore, how can "signal diversity" be quantified? Is diversity used in the same sense as complexity? If that is the case, what definition of complexity is being used by the authors? The review paper by Tononi et al. (1998) presents different kinds of complexity. Does "signal diversity" correspond to one of those types of complexity?

- Significant differences (figure 2) are reported using Lempel-Ziv complexity (LZc), with authors claiming that these reflect changes to "signal diversity". Have the authors checked the spectra of the signals and evaluated any possible differences using linear methods based on the Fourier transform? In some cases LZc might be only reflecting changes that can be observed with spectral methods. I would ask the authors to use surrogate data analysis to test the validity of their results in the context of complexity changes.

- The authors claim that "PCI is a measure of the global, spatio-temporal complexity of evoked, cortical responses to a local perturbation". Given that EEG signals are heavily affected by volume conduction, would this affect PCI results? This should be discussed.

- Line 114: The authors mention that LZc, amplitude coalition entropy (ACE), and synchrony coalition entropy (SCE) capture how "disordered" a signal is. Is this really the case? Do LZc, ACE and SCE return the highest values when analysing signals showing complete disorder (i.e. noise)? This needs to be investigated and discussed.

- Line 156: The aim of the study is stated, but what was the research hypothesis?

- Line 326: How many data samples were available for the calculation of LZc? Have the authors checked that the number of data points is enough for a stable and reliable estimate of the LZc of such a short signal?

- Line 359: Why was the mean and not the median used as the threshold for the calculation of LZc and ACE? The median is much more robust to any outliers in the data. I would recommend comparing the results obtained with the mean to those that would be obtained using the median instead.

Minor points:

- Line 78: Please give some details on the recent discoveries mentioned in this sentence.

- Line 184: Please include details about the age and sex of the participants in the study. EEG complexity changes with age and/or sex.

- Line 299: Who was responsible for the visual inspection of the signals and selection of those for further analysis? A trained expert?

- Line 307 to 309: Please include graphical examples of components with artefacts.

- Lines 314 and 342: What were the orders of the filters?

- Line 315: Why were the signals down-sampled to a sampling frequency different to the one used in the spontaneous EEG recordings?

- Line 339: Why were artefactual channels interpolated? Explain what you mean by this.

- Line 362: What was the rationale for only analysing 9 channels out of 62 instead of 10? The explanation provided is not clear enough.

- Lines 367 and 368: What was the rationale for using the normalisation detailed therein instead of the usual one applied when computing LZc from a signal?

- Lines 517 and 518: How much of it is integration and how much is just merely a volume conduction effect?

- Lines 580 and 581: Does it make sense to evaluate correlation with such a small sample size?

- Lines 632 and 649: "TEP's" should read "TEPs". The apostrophe should be used to indicate either possession or the omission of letters or numbers, but not to create plural forms of acronyms.

- The spelling used is inconsistent, mixing British English and American English. I would advise using the former, correcting mistakes like "center" (should read "centre"), "analog" (should read "analogue"), "setup" (should read "set-up"), etc.

- "Data" should be treated as a plural noun in scientific contexts (i.e. "data was" should read "data were"). Please correct this throughout the manuscript.

References:

Tononi et al. (1998) Complexity and coherency: integrating information in the brain, Trends in Cognitive Sciences, Vol. 2, No. 12, 474-484.

Reviewer #2: In this paper, the authors addressed how EEG indices of differentiation and integration are altered by sub-anaesthetic levels of ketamine compared to normal wakefulness. They showed that perturbational complexity index (PCI), which assesses differentiation as well as integration, is not significantly different between sub-anaesthetic levels of ketamine and normal wakefulness. By contrast, Lempel Ziv complexity (LZc), amplitude coalition entropy (ACE), or synchrony coalition entropy (SCE), which assesses differentiation alone, is higher in sub-anaesthetic levels of ketamine compared to normal wakefulness. The authors concluded that these indices reflect distinct aspects of consciousness. Specifically, they proposed that PCI reflects the brain’s ability to sustain consciousness, whereas LZc, ACE, or SCE reflects the richness of conscious contents.

Overall this is a solid paper. However, there are places where the lack of methodological details has rendered the conclusions / interpretations unjustified. The paper will benefit from a minor revision that clarifies the following points:

1. Disentangle the effect of analysis from the effect of data: The perturbational complexity index (PCI) is calculated from TSM-evoked EEG activity. By contrast, the Lempel Ziv complexity (LZc), amplitude coalition entropy (ACE), or synchrony coalition entropy (SCE) is calculated from spontaneous EEG activity. Therefore, it is unclear whether the differences between these indices in their changes (or lack of changes) from normal wakefulness to sub-anaesthetic levels of ketamine are due to the data themselves being differently affected by sub-anaesthetic levels of ketamine, or are instead due to the analyses / indices being differently affected by sub-anaesthetic levels of ketamine. It will be nice if the authors can provide some standard analyses which can be applied to both the TMS-evoked EEG activity and the spontaneous EEG activity and assess whether the data themselves are differently affected by sub-anaesthetic levels of ketamine. Or if performing such analyses will be difficult, it will be nice if the authors can provide some discussion on this issue.

2. Lack of experimental details: The authors proposed that the increase in Lempel Ziv complexity (LZc), amplitude coalition entropy (ACE), or synchrony coalition entropy (SCE) from normal wakefulness to sub-anaesthetic levels of ketamine reflects the enriched conscious contents induced by ketamine. As part of the argument, they show that Lempel Ziv complexity (LZc), amplitude coalition entropy (ACE), or synchrony coalition entropy (SCE) is higher under eye open condition compared to eye close condition. However, the authors did not explain whether the eye open data were collected in a fully dark room (which are often the case for visual experiments), or in a normal room. If the eye open data were collected in a fully dark room, it is possible that the visual experiences would be richer under the eye close condition compared to the eye open condition, as closing the eye may facilitate visual imagery. If indeed so, this would contradict the authors’ argument. On a related note, the author did not examine whether the impacts of experimental condition (eye close versus eye open) on Lempel Ziv complexity (LZc), amplitude coalition entropy (ACE), or synchrony coalition entropy (SCE), differs significantly between sub-anaesthetic levels of ketamine and normal wakefulness. If the visual experiences under sub-anaesthetic levels of ketamine are largely induced by top-down visual imagery, as opposed to bottom-up visual inputs from environment, having the eyes closed versus open would have weaker impacts here, compared to the impacts they produced during normal wakefulness.

3. Asymmetry in experimental condition: In this study, perturbational complexity index (PCI) / TMS-evoked EEG data were acquired under the experimental condition of eye open; by contrast, Lempel Ziv complexity (LZc), amplitude coalition entropy (ACE), or synchrony coalition entropy (SCE) / spontaneous EEG data were collected under the experimental condition of eye open as well as the experimental condition of eye close. It would benefit the readers if the authors could provide rationales behind such experimental design: namely, why there lacked the eye close condition for perturbational complexity index (PCI) / TMS-evoked EEG data.

6. PLOS authors have the option to publish the peer review history of their article (what does this mean?). If published, this will include your full peer review and any attached files.

Reviewer #1: No

Reviewer #2: No

---

## [Author Response · Author response to Decision Letter 0]

12 Oct 2020

We have no further responses to the editor or reviewer comments than the uploaded document, and are looking forward to any replies or discussions to further the article in the submission process.

---

## [Editor Report · Decision Letter 1]

27 Oct 2020

Increased signal diversity/complexity of spontaneous EEG, but not evoked EEG responses, in ketamine-induced psychedelic state in humans

PONE-D-20-19579R1

Dear Dr. Farnes,

We’re pleased to inform you that your manuscript has been judged scientifically suitable for publication and will be formally accepted for publication once it meets all outstanding technical requirements.

Kind regards,

Vladyslav Vyazovskiy, PhD

Academic Editor

PLOS ONE
---

## [Editor Report · Acceptance letter]

12 Nov 2020

PONE-D-20-19579R1 

Increased signal diversity/complexity of spontaneous EEG, but not evoked EEG responses, in ketamine-induced psychedelic state in humans 

Dear Dr. Farnes:

I'm pleased to inform you that your manuscript has been deemed suitable for publication in PLOS ONE. Congratulations! Your manuscript is now with our production department. 

Kind regards, 

on behalf of

Dr. Vladyslav Vyazovskiy 

Academic Editor

PLOS ONE